# Evaluation of 16S rDNA Heart Tissue PCR as a Complement to Blood Cultures for the Routine Etiological Diagnosis of Infective Endocarditis

**DOI:** 10.3390/diagnostics11081372

**Published:** 2021-07-30

**Authors:** Raquel Rodríguez-García, María Ángeles Rodríguez-Esteban, Jonathan Fernández-Suárez, Ana Morilla, Enrique García-Carús, Mauricio Telenti, Carlos Morales, Guillermo Muñiz Albaiceta, Javier Fernández

**Affiliations:** 1Unidad de Cuidados Intensivos Cardiológicos, Hospital Universitario Central de Asturias, 33011 Oviedo, Spain; rakel_20r@hotmail.com (R.R.-G.); angelesresteban@gmail.com (M.Á.R.-E.); gma@crit-lab.org (G.M.A.); 2Instituto de Investigación Sanitaria del Principado de Asturias (ISPA), 33011 Oviedo, Spain; 3CIBER-Enfermedades Respiratorias, Instituto de Salud Carlos III, 28220 Madrid, Spain; 4Servicio de Microbiología Clínica, Hospital Universitario Central de Asturias, 33011 Oviedo, Spain; jofersua@hotmail.com (J.F.-S.); anamorillamorilla@gmail.com (A.M.); mauritelenti@gmail.com (M.T.); 5Servicio de Medicina Interna, Hospital Universitario Central de Asturias, 33011 Oviedo, Spain; e.g.carus@gmail.com; 6Servicio de Cirugía Cardiaca, Hospital Universitario Central de Asturias, 33011 Oviedo, Spain; motocarlos24@yahoo.com; 7Departamento de Biología Funcional, Instituto Universitario de Oncología del Principado de Asturias, Universidad de Oviedo, 33003 Oviedo, Spain

**Keywords:** infective endocarditis, 16S rDNA PCR, blood culture, valve heart culture, molecular diagnosis

## Abstract

Identification of the causative pathogen is required to optimize the effective therapy in infective endocarditis (IE). The aim of this study was to assess a 16S rDNA PCR to identify bacteria from heart valve tissues and to evaluate its usefulness as a complement to blood and removed valves cultures. A total of 266 patients diagnosed with IE from January 2015 to December 2019 were evaluated. Results between 16S rDNA PCR from heart valve tissues were compared with microbiological cultures. Blood cultures were positive in 83.5% of patients diagnosed with IE, while 39.6% and 71.8% of the evaluated heart valve samples were positive by culture and 16S rDNA PCR, respectively. For 32 (12%) patients, 16S rDNA tissue PCR provided valuable information supporting the results of blood cultures in the case of bacteria characteristic from the skin microbiota. Additionally, a microorganism was identified by using 16S rDNA PCR in 36% of blood culture-negative cases. The present study reveals that molecular diagnosis using 16S rDNA tissue PCR provides complementary information for the diagnosis of IE, and it should be recommended in surgical endocarditis, especially when blood cultures are negative.

## 1. Introduction

Infective endocarditis (IE) is a serious disease associated with the infection of a native or prosthetic heart valve or an implanted cardiac device [1,2]. Appropriate recognition and management of this infection remain a clinical challenge. The outcomes of affected patients are still poor, with mortality rates of up to 30% [3]. Currently, blood and heart valve tissue cultures are the gold standards for the identification of the causative pathogen, which is important for the initiation of an optimal targeted antimicrobial therapy. However, up to 10% of all IE are associated with negative blood cultures [4,5]. These are related to cases in which antibiotic therapy is initiated before sampling or with IE caused by intracellular, fastidious, slow-growing, and/or unculturable microorganisms such as *Coxiella burnetii* or *Tropheryma whipplei* [6]. Consequently, to achieve a precise diagnosis, current guidelines recommend alternative methods for pathogen detection [7].

In recent years, molecular detection methods have become an important part of IE diagnosis and represent an alternative approach, especially when the culture is negative. Amplification through polymerase chain reaction (PCR) and further sequencing of variable regions of the 16S rDNA gene is a widely accepted method for identifying bacteria [8,9]. There is growing evidence supporting the use of this approach for the detection of bacteria from removed heart valves with sensitivities around 95% [10]. Therefore, 16S rDNA valve PCR has been established as an independent, specific, and rapid method to assist pathogen detection in cases of blood and valve negative cultures and can also be useful to confirm previously detected microorganisms [11].

The aim of this study was to assess a 16S rDNA PCR method to identify bacteria from heart valve tissues in a large series of patients and to evaluate the usefulness of this technique as a complement to the culture of the valve and blood for the etiological diagnosis of IE.

## 2. Materials and Methods

### 2.1. Patients

All patients admitted to a tertiary hospital in northern Spain (Hospital Universitario Central de Asturias, Oviedo, Spain) with a diagnosis of IE from January 2015 to December 2019 were included in the study. Our institution is a 1039-bed university hospital and has a cardiovascular surgery department that provides patient care to a population of 1 million people. The diagnoses of IE were established based on clinical, microbiological, and echocardiographic findings according to the modified Duke’s criteria [12]. Patients’ clinical records were retrospectively reviewed using a standardized data collection sheet that included clinical and microbiological information such as location of IE, type of valve tissue, and blood and valve culture results.

### 2.2. Microbiological Cultures

Microbiology tests, including microbiological culture and bacterial identification, were carried out at the hospital Clinical Microbiology Unit. Blood cultures were performed on each patient and were inoculated in two separate sets of Standard Aerobic (SA) and Standard Anaerobic (SN) BacT/ALERT^®®^ blood culture bottles (bioMérieux Inc., Durham, NC, USA) which were processed in a BacT/ALERT^®®^ VIRTUO™ machine (VIRTUO™, bioMérieux Inc., Hazelwood, MO, USA). Blood cultures bottles were incubated 15 days, and, if positive, standard subcultures were performed according to IDSA guidelines [13]. Bacterial identification of growing colonies was performed by MALDI-TOF MS (Bruker Daltonik, Bremen, Germany). Heart valve samples (native and prosthetic), pacemaker catheter electrodes, and/or heart tissue as vegetations or perivalvular abscess were collected under sterile conditions in patients who underwent surgery and were processed for bacterial and fungal culture according to IDSA guidelines [13].

### 2.3. 16S rDNA PCR Assays

Additionally, heart samples were evaluated by a 16S rDNA-based PCR and further sequencing except for those cases in which there was an unequivocal diagnosis by culture. Bacterial DNA was extracted using MagCore (RBC Bioscience, New Taipei City, Taiwan). A PCR amplification of the 16S RNA-encoding gene and subsequent sequencing was performed as previously described [14]. Sequence results were compared with data from two different databases (Blast—Basic Local Alignment Search Tool and leBIBI—Quick Bioinformatic Phylogeny of Prokaryotes—v 1.1).

### 2.4. Data Analysis

Data were analyzed using SPSS software (IBM SPSS Statistics 19). Descriptive statistics were computed for all study variables. Differences between the length antimicrobial treatment groups were performed using Student’s *t*-test. For all comparisons, a *p* value of less than 0.05 was considered significant.

### 2.5. Ethics

Ethical approval for the study was provided by the Research Ethics Committee of the Principality of Asturias (reference CEImPA-2018-210, in 14 August 2018).

## 3. Results

### 3.1. Patient Demographics and Clinical Features

A total of 266 patients with a diagnosis of IE according to the modified Duke’s criteria were included in the study. Demographic characteristics and clinical features are displayed in Table 1. Most patients were male (*n* = 171, 64.3%) with a mean age of 68 years (range 6–95). Native valve endocarditis accounted for 140 (52.6%) episodes, while prosthetic valve endocarditis accounted for 115 (43.2%). All patients were treated with antibiotics before valve surgery. Valve replacement was carried out in 154 patients.

### 3.2. Previous Antimicrobial Therapy and Its Effect on Microbiological Results

The mean duration of antimicrobial treatment before surgery was 10.3 days (range: 1–105). In order to analyze the potential effect of antimicrobial therapy on microbiological results, results from heart valves and other tissues were analyzed in relation to the length of antimicrobial treatment before surgery (Figure 1). 16S rDNA valve PCR was positive in 71.4% of patients who received antibiotics for ≥10 days before surgery, while only 24.1% of the same patients had a positive valve culture. The mean duration of preoperative treatment was similar in patients with negative and positive PCR (10.7 ± SD and 11.5 ± SD days, respectively, *p* = 0.8). However, the mean duration of antibiotic treatment before surgery was longer in patients with negative valve culture (11.2 ± SD and 6.7 ± SD days for those with negative and positive valve culture, respectively, *p* = 0.005). These data show that bacterial DNA could persist during antibiotic treatment in infected valve tissue in contrast to the valve culture, wherein the latter’s results are affected by the length of the antimicrobial therapy.

### 3.3. Microbiological Methods Results

A causative microorganism was identified in most of the patients. No microorganism was detected neither by blood culture, 16S rDNA PCR, or culture of valvular specimen for only 18 subjects. However, all these episodes were clinically classified as possible IE based on clinical and echocardiographic findings according to the modified Duke’s criteria [12].

Blood cultures were performed on the 266 patients. In addition, valve culture was completed in 154. The 16S rDNA PCR was carried out in 103 out of these 154 cases. Microbiological test performances were as follows: 222 patients (222/266, 83.5%) had positive blood cultures, 74 (74/103, 71.8%) had a positive result for 16S rDNA valve PCR, and 61 (61/154, 39.6%) yielded a positive result of valve culture. Microbiological results are summarized in Figure 2.

Microbiological test results for each individual patient and results grouped by species are detailed in Appendix A and Table 2, respectively. The main etiological agents causing IE in our series were coagulase-negative staphylococci (*n* = 69) followed by viridans group streptococci (*n* = 49), *Staphylococcus aureus* (*n* = 48) and enterococci (*n* = 46).

### 3.4. Concordances and Discrepancies between the Different Evaluated Methods

Concordances and discrepancies between the different evaluated microbiological methods are shown in Figure 3 and Figure 4. When both blood culture and 16S rDNA valve PCR were positive, results were concordant in all but two cases (Table 3). One of them corresponded to a case of IE by *Candida*
*albicans* and *Streptococcus dysgalactiae*. As 16S rDNA PCR only detects bacterial DNA, the result of the valve analysis was expected to be negative. However, *S. dysgalactiae* was identified, and the episode was considered and treated as a polymicrobial infection. The second case corresponded to mixed endocarditis caused by *S. epidermidis* and *Granullicatella adiacens*, detected by 16S rDNA PCR and blood culture, respectively. As both germs are known etiological agents of infective endocarditis, it could not be ruled out that the infection was caused by both germs, and treatment was targeted to both pathogens.

Regarding valve cultures, several discrepancies were found (Table 3). These discrepancies were mostly considered as contaminations of valve cultures, as microorganisms detected by the other evaluated methods were more plausible.

### 3.5. Diagnostic Benefit from Valve PCR

16S rDNA valve PCR contributed with clinically relevant information, allowing an etiological diagnosis in 16 cases of IE with valve and blood culture-negative results (Table 4). Moreover, this was the only diagnostic method that detected species such as *C. burnetii* and *T. whipplei*. Additionally, in 16 other cases, 16S rDNA PCR supported the results of blood and/or valve cultures in case of IE caused by coagulase-negative staphylococci or other bacteria from the skin microbiota, such as *Cutibacterium acnes* (Table 4).

## 4. Discussion

The present study evaluates the use of 16S rDNA heart tissue PCR combined with culture-based methods for the identification of the etiological microorganism in IE. Our results confirm that blood culture-negative endocarditis (BCNE) benefits by adding microbiological molecular assays [7]. In addition, the present work also confirms previous evidence that 16S rDNA heart valve PCR testing is particularly useful in comparison to tissue culture in patients undergoing surgery [15] and to corroborate microorganisms that can sometimes be considered contaminants.

A total of 154 (57.9%) patients in this series required surgical treatment of IE. Although preoperative blood cultures detected the etiological agent in almost two-thirds of the cases, 16S rDNA valve PCR analysis and/or valve tissue culture contributed to the microbiologic diagnosis in 9.8% (26/266) of patients. The test performance was 39.6% (61/154) for valve culture and 71.8% (74/103) for 16S rDNA valve PCR. Other studies have reported performances of 46% and 92% for the same methods, respectively [16], in line with our results. Due to the high performance of molecular methods in comparison to culture, some studies have questioned the need for routine culture of heart valves when 16S rDNA PCR analysis is available [17], and current evidence suggests that 16S rDNA PCR should be preferentially conducted after a blood test to identify the causative pathogen. However, applying only molecular methods to valves cannot be recommended, as the etiological agent is not recovered and therefore, the performance of antimicrobial susceptibility testing to guide the instauration of an optimal targeted antimicrobial therapy is not possible. Therefore, culture-based diagnostic techniques are an essential part of IE management.

Our series has a proportion of BCNE of 16.5% (44/266). BCNE rates are highly variable in the literature, ranging from 2.5 to 31% [18]. This variability is conditioned by several factors, such as antimicrobial treatments prior to culture and/or the incidence of non-cultivable or fastidious microorganisms [19]. Molecular analysis of heart valves was found to be the only method to establish the etiology of BCNE, giving consistent results in 16 out of 44 patients (36%). These results agree with recent articles where 16S rDNA valve PCR was positive in 41% of surgical blood culture-negative endocarditis [20].

Most of the patients analyzed in the present work were under antibiotic therapy at the time of heart valve excision. Although this is the usual clinical practice, only a few studies have documented the average length of antibiotic treatment given prior to surgery. The mean duration of previous antimicrobial treatment in our series was similar in patients with negative and positive 16S rDNA valve PCR, respectively. However, we found a significant difference in the duration of antibiotic treatment before surgery between patients with negative and positive valve cultures. Previous works reported that the sensitivity of 16S rDNA valve PCR is independent of the length of antibiotic treatment before surgery, in contrast to culture sensitivity [21,22]. This can be explained by the persistence of DNA in spite of antibiotic treatment that reduces microorganism viability and consequently impairs their growth in microbiological cultures [23].

A potentially cultivable microorganism was detected by 16S rDNA in seven cases of BCNE from our series. In four of them, *Streptococcus* spp. was identified, and three were positive for *Staphylococcus* spp. Similar studies reported other potentially cultivable microorganisms such as enterococci and nutritionally variant streptococci in BCNE [7]. Additionally, 16S rDNA valve PCR significantly increases the ability to detect intracellular organisms such as *Coxiella* spp., *Tropheryma* spp., and *Bartonella* spp. [24]. In the present work, we reported four, two, and one cases of *C. burnetti*, *T. whipplei,* and *B. henselae* IE, respectively, all detected by 16S rDNA valve PCR. Even though the epidemiology of these pathogens has regional differences [15], underdiagnosis has been demonstrated in southern European countries after the incorporation of new molecular tools [25]. In a series of 1383 patients in France, *C. burnetii* accounted for at least 5% of all IE [26]. According to several authors, the diagnosis is often delayed for several months [27], and the infection is involved in 37–45% of BCNE [28,29]. Additionally, in a recent multicenter study carried out in Spain, 28.2% of BCNE reported were caused by *Coxiella burnetii* and 10.3% by *Tropheryma whipplei* [30]. These are all emerging pathogens, perhaps not because of the increase in cases but because of the improvement in diagnostic techniques that allow their detection.

Furthermore, the performance of 16S rDNA PCR in our study was useful to interpret blood culture results, for instance, to confirm infections by bacteria from the skin microbiota, a benefit that has already been highlighted in other works [20].

Despite the advantages of valvular 16S rDNA PCR, it should be noted that this test was negative in 25 samples of patients with a positive blood culture. This was unexpected; however, a potential explanation is that PCR result is restricted by the site of the excised tissue, and infection is not distributed homogeneously in the valve [31].

Metagenomic sequencing methods are currently being developing to provide information on the etiological diagnosis and its antibiotic susceptibility profile in patients with IE [32], potentially offering a powerful method to deal with negative blood cultures, especially in patients who are not undergoing surgical treatment, and to deal with polymicrobial infections [31,33]. However, Santibáñez et al. recently reported that the use of a metagenomic approach in the study of IE in resected heart valves provided no additional significant information compared to conventional 16S rDNA PCR, except for polymicrobial IE [34].

The present study has limitations, such as its retrospective nature or the fact that 16S rDNA PCR was not performed on all valve samples. However, it is a very large series compared to other previous works published in the literature, and it serves to support the usefulness of this approach for the diagnosis and management of EI. These additive results contribute to establishing the advantages and limitations of emerging techniques.

## 5. Conclusions

Our results illustrate the role of 16S rDNA heart tissue PCR in the microbiological diagnosis of infective endocarditis. This technique may provide additional information which can guide antibiotic therapy, especially in patients with negative blood cultures. Furthermore, this study supports the use of this method to confirm the results of patients with positive blood cultures suspected as contaminants, such as those cases caused by bacteria from the skin microbiota.

## Figures and Tables

**Figure 1 diagnostics-11-01372-f001:**
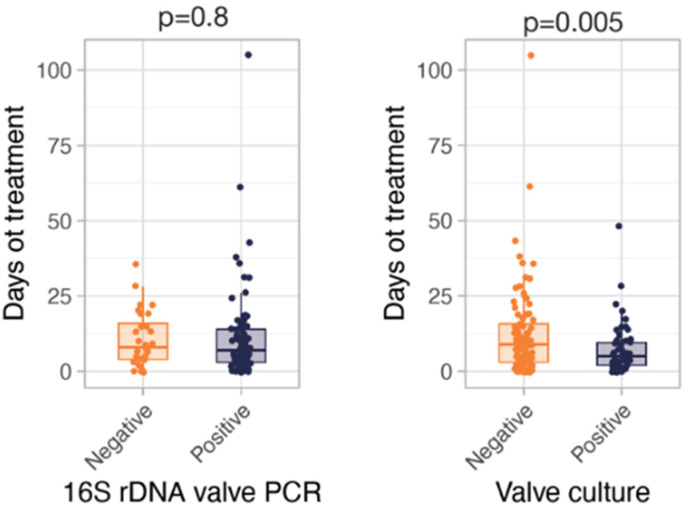
Antimicrobial therapy duration before surgery and its relatedness with valve culture and 16S rDNA PCR results. Bars represent the median of days.

**Figure 2 diagnostics-11-01372-f002:**
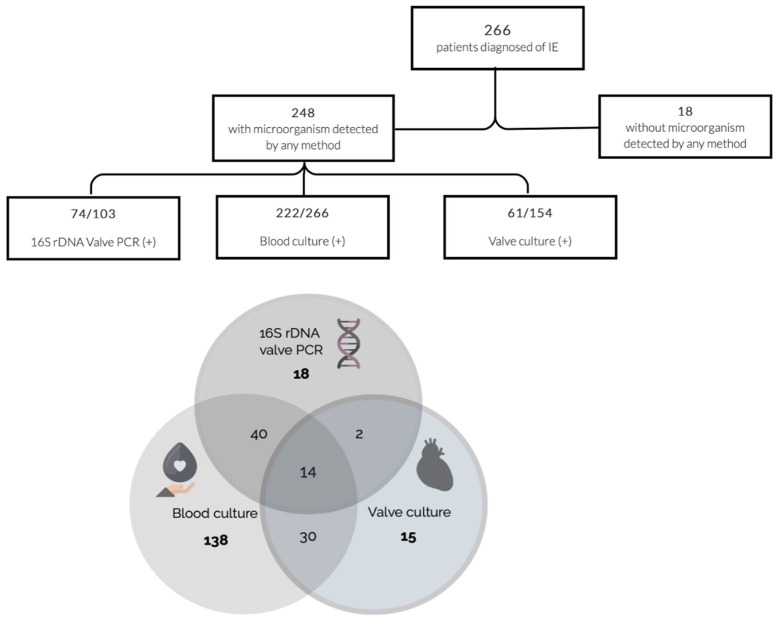
Microorganism detection by the different microbiological methods evaluated (blood culture, heart valve or vegetation culture, and/or 16S rDNA PCR).

**Figure 3 diagnostics-11-01372-f003:**
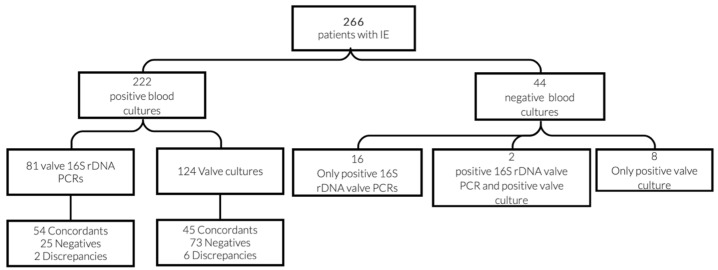
Concordance between blood and heart valve culture and/or 16S rDNA valve PCR results.

**Figure 4 diagnostics-11-01372-f004:**
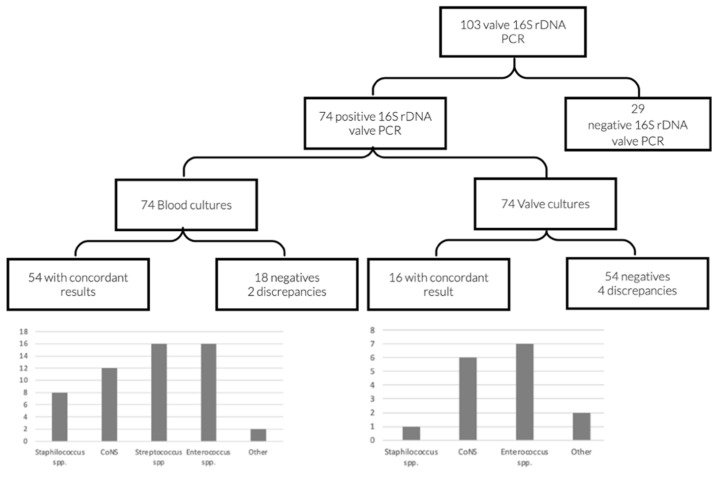
Distribution of microorganisms with concordant results between blood and valve culture in comparison with 16S rDNA valve PCR.

**Table 1 diagnostics-11-01372-t001:** Clinical features of patients with infective endocarditis included in the study.

Demographics	*n* (%)
Male sex	171 (64.3)
Age mean ± SD (years)	68 ± 13
**Medical history**	
Native valve	140 (52.6)
Prosthetic valve	115 (43.2)
Intracardiac device	18 (6.8)
Previous endocarditis	20 (7.5)
**Location**	
Aortic	138 (51.9)
Mitral	65 (24.4)
Pulmonary	2 (0.8)
Tricuspid	3 (1.1)
Pacemaker	15 (5.6)
Multiple valves	40 (15.1)
Others	3 (1.1)
**Clinical course**	
Surgical treatment	154 (57.9)
Time of antimicrobial treatment before surgery (mean ± SD, days)	10.3 ± 12.8
Time of antimicrobial treatment after surgery (mean ± SD, days) *	42.3 ± 19.5
Time of total antimicrobial treatment (mean ± SD, days) *	42.7 ± 22.5
Deaths	72 (27.1)

SD, standard desviation. * Patients on suppressive treatment for more than 90 days have been removed.

**Table 2 diagnostics-11-01372-t002:** Microbiological results grouped by species of patients diagnosed with infective endocarditis.

		Number of Microorganisms Detected						
Bacterial Species	Number of Cases	Only Positive Blood Culture	Only Positive Heart Tissue Culture	Only Positive Heart Tissue 16S rDNA PCR	Positive Blood Culture	Positive Heart Tissue Culture	Positive Heart Tissue 16S rDNA PCR	Positive in All Methods
***Staphylococcus aureus***	**48**	**32**	**2**	**1**	**45**	**9**	**8**	**1**
**Coagulase-negative staphylococci (CoNS)**	**69**	**41**	**3**	**2**	**64**	**19**	**15**	**6**
*Staphylococcus epidermidis*	56	33	1	1	54	16	13	6
*Staphylococcus lugdunensis*	5	2	0	1	4	1	2	0
*Staphylococcus auricularis*	2	1	1	0	1	1	0	0
*Staphylococcus hominis*	2	2	0	0	2	0	0	0
*Staphylococcus haemolyticus*	1	0	1	0	0	1	0	0
*Staphylococcus warneri*	1	0	0	0	1	1	0	0
Other CoNS	2	2	0	0	2	0	0	0
**Viridans group streptococci**	**49**	**28**	**0**	**3**	**46**	**3**	**18**	**0**
*Streptococcus mitis*	3	0	0	2	1	0	3	0
*Streptococcus mutans*	3	0	0	1	2	0	3	0
*Streptococcus anginosus*	2	1	0	0	2	1	0	0
*Streptococcus constellatus*	1	1	0	0	1	0	0	0
*Streptococcus gallolyticus*	17	10	0	0	17	2	5	0
*Streptococcus gordonii*	6	4	0	0	6	0	2	0
*Streptococcus oralis*	6	3	0	0	6	0	3	0
*Streptococcus parasanguis*	2	2	0	0	2	0	0	0
*Streptococcus salivarius*	1	1	0	0	1	0	0	0
*Streptococcus sanguinis*	6	4	0	0	6	0	2	0
Other	2	2	0	0	2	0	0	0
**beta-hemolytic streptococci**	**5**	**2**	**0**	**1**	**4**	**1**	**2**	**0**
*Streptococcus dysgalactiae*	1	0	0	0	1	1	0	0
*Streptococcus agalactiae*	4	2	0	1	3	0	2	0
***Enterobacterales* and nonfermenting Gram-negative bacilli**	**8**	**3**	**0**	**2**	**6**	**2**	**3**	**0**
*Escherichia coli*	3	1	0	1	2	0	2	0
*Enterobacter cloacae*	1	1	0	0	1	0	0	0
*Chryseobacterium* spp.	1	0	0	1	0	0	1	0
*Pseudomonas aeruginosa*	2	1	0	0	2	1	0	0
*Serratia marcescens*	1	0	0	0	1	1	0	0
**Enterococci**	**46**	**24**	**1**	**0**	**47**	**15**	**16**	**7**
*Enterococcus faecalis*	43	21	1	0	42	14	15	7
*Enterococcus faecium*	3	2	0	0	3	0	1	0
***Candida* spp.**	**2**	**1**	**0**	**0**	**2**	**1**	**0**	**0**
**Other**	**17**	**6**	**1**	**7**	**7**	**3**	**10**	**0**
*Abiotrophia defectiva*	1	1	0	0	1	0	0	0
*Aerococcus urinae*	1	1	0	0	1	0	0	0
*Agregatibacter actinomycetemcomitans*	2	2	0	0	2	0	0	0
*Listeria monocytogenes*	1	1	0	0	1	0	0	0
*Tropheryma whipplei*	2	0	0	2	0	0	2	0
*Coxiella burnetii*	4	0	0	4	0	0	4	0
*Cutibacterium acnes*	4	0	1	0	1	3	3	0
*Bartonella henselae*	1	0	0	1	0	0	1	0
*Streptococcus pneumoniae*	1	1	0	0	1	0	0	0
**Polymicrobial infection**	**4**	**0**	**1**	**0**	**3**	**3**	**2**	**0**

**Table 3 diagnostics-11-01372-t003:** Discrepancies between the different evaluated methods for the diagnosis of infective endocarditis.

**Blood Cultures**	**Heart Valve or Vegetation Cultures**	**16S rDNA Heart Valves PCR**
**Discrepancies between blood cultures and 16S rDNA heart valve PCR**		
*Candida albicans*	*Candida albicans*	*Streptococcus dysgalactiae*
*Granulicatella adiacens*	Negative	*Staphylococcus epidermidis*
**Discrepancies between heart valve or vegetation cultures and the others**		
*Streptococcus oralis*	*Cutibacterium acnes*	*Streptococcus oralis*
*Staphylococcus warneri*	*Staphylococcus warneri* and *Cutibacterium acnes*	Negative
*Streptococcus dysgalactiae*	*Streptococcus dysgalactiae* and *Staphylococcus aureus*	Not performed
*Agregatibacter actinomycetemcomitans*	*Klebsiella pneumoniae*	Not performed
*Enterococcus faecalis*	*Staphylococcus cohnii*	*Enterococcus faecalis*
*Staphylococcus aureus*	*Cutibacterium acnes*	Negative
*Streptococcus mutans*	*Staphylococcus haemolyticus*	*Streptococcus mutans*

**Table 4 diagnostics-11-01372-t004:** Diagnostic benefit of 16S rDNA PCR for patients with infective endocarditis.

Blood Cultures	Heart Valve or Vegetation Cultures	16S rDNA Heart Valves PCR	Number of Cases
Patients with only positive valve PCR			
Negative	Negative	*Coxiella burnetii*	4
Negative	Negative	*Tropheryma whipplei*	2
Negative	Negative	*Streptococcus mitis*	2
Negative	Negative	*Staphylococcus aureus*	1
Negative	Negative	*Staphylococcus epidermidis*	1
Negative	Negative	*Escherichia coli*	1
Negative	Negative	*Staphylococcus lugdunensis*	1
Negative	Negative	*Streptococcus agalactiae*	1
Negative	Negative	*Bartonella henselae*	1
Negative	Negative	*Streptococcus mutans*	1
Negative	Negative	*Chyseobacterium* spp.	1
**Patients with coagulase-negative staphylococci and skin commensals in blood cultures that are confirmed by concordant valve PCR**			
*Staphylococcus epidermidis*	*Staphylococcus epidermidis*	*Staphylococcus epidermidis*	6
*Staphylococcus epidermidis*	Negative	*Staphylococcus epidermidis*	6
*Staphylococcus lugdunensis*	Negative	*Staphylococcus lugdunensis*	1
Negative	*Cutibacterium acnes*	*Cutibacterium acnes*	2
*Cutibacterium acnes*	Negative	*Cutibacterium acnes*	1

## Data Availability

Data are contained within the article.

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
