# Peer review of "Evaluation of 16S rDNA Heart Tissue PCR as a Complement to Blood Cultures for the Routine Etiological Diagnosis of Infective Endocarditis"

_diagnostics, 2021, doi:10.3390/diagnostics11081372_

Round 1

Reviewer 1 Report

The manuscript described the use of the 16S assay for identification of possible causes of endocarditis. The manuscript is well written and the data is well presented, but the method is not new. Comparison between various molecular assays might be more interesting for the reader and of added value. Since the ultimate aim of the Journal "diagnostics" to induce novel diagnostics for the scientific community, the current study is not in that direction. 

Author Response

General Comments from Reviewer 1:

Comment:

The manuscript described the use of the 16S assay for identification of possible causes of endocarditis. The manuscript is well written and the data is well presented, but the method is not new. Comparison between various molecular assays might be more interesting for the reader and of added value. Since the ultimate aim of the Journal "diagnostics" to induce novel diagnostics for the scientific community, the current study is not in that direction. 

Reply:

The authors would like to thank the Reviewer for their comments. Accordingly, throughout the manuscript, we have revised and verified there are few articles that evaluate the usefulness of molecular methods such as 16S rDNA PCR for the etiological diagnosis of infective endocarditis.

Several examples were previously articles such as Min-Sun et al, Miller et al, Vondracek et al and recently Armstrong et al who evaluated 80, 68, 57 and 146 cases respectively.

Therefore, from our perspective, the strength of our study is reporting the experience from a large number of patients (266 patients) with infective endocarditis in a tertiary hospital that has accrued experience in the development and implementation of complex user-designed molecular assays. After all, it accurately reflects the routine utility of 16S rDNA PCR.

In addition, we carried out an evaluation of the antibiotic treatment influence on valve culture and 16S rDNA PCR results.

Reviewer 2 Report

The authors confirmed in this study that 16S rDNA heart tissue PCR is a complementary method to blood culture assay for endocarditis diagnosis. PCR assay has been proposed to solve the detection problem of the blood culture-negative infective endocarditis. Slow growing and fastidious microorganisms, such as Coxiella burnetii and Tropheryma whipplei, are difficult to culture, thus they are not detected with the standard blood culture assay. On the other hand, PCR assay can detect minor pathogens with its high detection sensitivity although some discrepancies exist among methods. The findings in this study are mainly to confirm the usefulness of 16S rDNA heart tissue PCR method for diagnosis. The results provided from this study will be helpful for readers who need to diagnose infective endocarditis.

Major comment:

  1. Although PCR is beneficial to detect minor pathogen, valve culture test seems to tell us different information for endocarditis diagnosis. In actual diagnosis, which method should be conducted first after blood test to identify the causative pathogen, 16S rDNA PCR or valve test? Like Figure 3, blood test negative cultures should be evaluated further to detect hidden causes. Can we conclude blood test with PCR test is informative enough to avoid routine valve test? Or still valve culture is needed?

Minor comments:

  1. Figure 1: This graph needs to improve for readers to understand the authors’ points. What are the numbers above the box-whisker plot? What is the difference between asterisk and rounds? There is a significant difference between -ve and +ve in valve culture, but in valve PCR. Then, what does this result indicate? I suppose that for antimicrobial therapy the authors would like to say that PCR method is more sensitive in the situation that pathogens are less after >10 days than valve culture method. More explanatory sentences are needed for readers in this section.
  2. Table 3, line 149-150: “When both blood culture and 16S rDNA valve PCR were positive, results were…(Table 3).” How to read Table3? Why are listed discrepancies in Table3 separated by a centre line? How different is top category from bottom category?
  3. Materials & methods: I suggest dividing this section with subheading like sampling, microbiology test, statistical analysis, and so on.
  4. Introduction and Discussion: The discussion section is redundant whereas the introduction section is compact.  
  5. There are several English errors.

Author Response

General Comments from Reviewer 2:

Comment:

The authors confirmed in this study that 16S rDNA heart tissue PCR is a complementary method to blood culture assay for endocarditis diagnosis. PCR assay has been proposed to solve the detection problem of the blood culture-negative infective endocarditis. Slow growing and fastidious microorganisms, such as Coxiella burnetii and Tropheryma whipplei, are difficult to culture, thus they are not detected with the standard blood culture assay. On the other hand, PCR assay can detect minor pathogens with its high detection sensitivity although some discrepancies exist among methods. The findings in this study are mainly to confirm the usefulness of 16S rDNA heart tissue PCR method for diagnosis. The results provided from this study will be helpful for readers who need to diagnose infective endocarditis.

Reply:

The authors would like to thank the Reviewer for their comments. Care has been taken to improve the work and address your concerns based on the specific comments below.

Major comment:

Although PCR is beneficial to detect minor pathogen, valve culture test seems to tell us different information for endocarditis diagnosis. In actual diagnosis, which method should be conducted first after blood test to identify the causative pathogen, 16S rDNA PCR or valve test? Like Figure 3, blood test negative cultures should be evaluated further to detect hidden causes. Can we conclude blood test with PCR test is informative enough to avoid routine valve test? Or still valve culture is needed?

Reply:

As noted by the Reviewer, our study demonstrated 16S rDNA PCR has more sensitivity than valve culture for the etiological diagnoses of infective endocarditis. However, the valve culture could confirm the diagnoses in some cases and, moreover, the isolation of the etiological agent remains essential for the performance of antimicrobial susceptibility testing to guide an optimal antimicrobial treatment.

It should be noted that molecular methods are not available in all microbiology services, unlike the accessibility and price of valve cultures. Therefore, we consider that routine heart valve cultures should continue to be performed until we could optimize and generalize molecular tests in Clinical Microbiology Units of all hospitals.

Following the editor's recommendations, we have modified the text as follows.

“Due to the high performance of molecular methods in comparison to culture, some studies have questioned the need for routine culture of heart valves when 16S rDNA PCR analysis is available [17], and current evidence suggests that 16S rDNA PCR should be preferently conducted after blood test to identify the causative pathogen. However, applying only molecular methods to valves cannot be recommended, as the etiological agent is not recovered and therefore performance of antimicrobial susceptibility testing to guide the instauration of an optimal targeted antimicrobial therapy is not possible. Therefore, culture-based diagnostic techniques are an essential part of IE management.”

Minor comments:

  1. Figure 1: This graph needs to improve for readers to understand the authors’ points. What are the numbers above the box-whisker plot? What is the difference between asterisk and rounds? There is a significant difference between -ve and +ve in valve culture, but in valve PCR. Then, what does this result indicate? I suppose that for antimicrobial therapy the authors would like to say that PCR method is more sensitive in the situation that pathogens are less after >10 days than valve culture method. More explanatory sentences are needed for readers in this section.

Reply:

Following the editor recommendations, we have modified Figure 1 as shown.

The text has been corrected as suggested. “These data show that bacterial DNA could persist during antibiotic treatment in infected valve tissue in contrast to the valve culture which results are affected by the length of the antimicrobial therapy”.

2. Table 3, line 149-150: “When both blood culture and 16S rDNA valve PCR were positive, results were…(Table 3).” How to read Table3? Why are listed discrepancies in Table3 separated by a centre line? How different is top category from bottom category?

Reply:

As noted by the Reviewer, changes have been made in Table 3 to make it easier to read.

Blood cultures

Heart valve or vegetation cultures

16S rDNA heart valves PCR

Discrepancies between blood cultures and 16S rDNA heart valve PCR

Candida albicans

Candida albicans

Streptococcus dysgalactiae

Granulicatella adiacens

Negative

Staphylococcus epidermidis

Discrepancies between heart valve or vegetations cultures and the others

Streptococcus oralis

Cutibacterium acnes

Streptococcus oralis

Staphylococcus warneri

Staphylococcus warneri and Cutibacterium acnes

Negative

Streptococcus dysgalactiae

Streptococcus dysgalactiae and Staphylococcus aureus

Not performed

Agregatibacter actinomycetemcomitans

Klebsiella pneumoniae

Not performed

Enterococcus faecalis

Staphylococcus cohnii

Enterococcus faecalis

Staphylococcus aureus

Cutibacterium acnes

Negative

Streptococcus mutans

Staphylococcus haemolyticus

Streptococcus mutans

3. Materials & methods: I suggest dividing this section with subheading like sampling, microbiology test, statistical analysis, and so on.

Reply:

Done according to recommendations.

4. Introduction and Discussion: The discussion section is redundant whereas the introduction section is compact.  

Reply:

Following the editor recommendations, we have modified the discussion section to avoid the redundance.

5. There are several English errors.

Reply:

We regret there were problems with the English. The article has been carefully revised by an English speaker to improve the grammar and readability.

Round 2

Reviewer 1 Report

Many thanks for the revised version. The authors have claimed that their study is unique because of the large number of patients included. In my point of view, this is not enough for publication, unless a multi-center was conducted.